# Phenotypical Characterization of C9ALS Patients from the Emilia Romagna Registry of ALS: A Retrospective Case–Control Study

**DOI:** 10.3390/genes16030309

**Published:** 2025-03-04

**Authors:** Andrea Ghezzi, Giulia Gianferrari, Elisa Baldassarri, Elisabetta Zucchi, Ilaria Martinelli, Veria Vacchiano, Luigi Bonan, Lucia Zinno, Andi Nuredini, Elena Canali, Matteo Gizzi, Emilio Terlizzi, Doriana Medici, Elisabetta Sette, Marco Currò Dossi, Simonetta Morresi, Mario Santangelo, Alberto Patuelli, Marco Longoni, Patrizia De Massis, Salvatore Ferro, Nicola Fini, Cecilia Simonini, Serena Carra, Giovanna Zamboni, Jessica Mandrioli

**Affiliations:** 1Department of Biomedical, Metabolic and Neural Sciences, University of Modena and Reggio Emilia, 41125 Modena, Italy; andreaghezzi140495@gmail.com (A.G.); elisa.baldassarri@studenti.unimore.it (E.B.); elibettizucchi@gmail.com (E.Z.); serena.carra@unimore.it (S.C.); giovanna.zamboni@unimore.it (G.Z.); jmandrio@unimore.it (J.M.); 2Neurology Unit, Azienda Ospedaliero Universitaria di Modena, 41126 Modena, Italy; martinelli.ilaria88@gmail.com (I.M.); fini.nicola@aou.mo.it (N.F.); ceciliasimonini24@gmail.com (C.S.); 3IRCCS Istituto delle Scienze Neurologiche di Bologna, Bellaria Hospital, 40139 Bologna, Italy; veriavacchiano@gmail.com; 4Dipartimento di Scienze Biomediche e Neuromotorie, University of Bologna, 40126 Bologna, Italy; luigi.bonan@studio.unibo.it; 5Department of Medicine and Surgery, University of Parma, 43121 Parma, Italy; lucia.zinno@gmail.com (L.Z.); andi.nuredini@unipr.it (A.N.); 6Neurology Unit, Arcispedale Santa Maria Nuova, AUSL-IRCCS Reggio Emilia, 42123 Reggio Emilia, Italy; elena.canali@asmn.re.it; 7Department of Neurology, Faenza and Ravenna Hospital, 48121 Ravenna, Italy; matteo.gizzi@auslromagna.it; 8Department of Neurology, G. Da Saliceto Hospital, 29121 Piacenza, Italy; e.terlizzi@ausl.pc.it; 9Department of Neurology, Fidenza Hospital, 43036 Fidenza, Italy; dmedici@ausl.pr.it; 10Department of Neuroscience and Rehabilitation, St. Anna Hospital, 44124 Ferrara, Italy; e.sette@ospfe.it; 11Department of Neurology, Infermi Hospital, 47923 Rimini, Italy; 12Department of Neurology and Stroke Unit, Bufalini Hospital, 47521 Cesena, Italy; simonetta.morresi@auslromagna.it; 13Department of Neurology, Carpi Hospital, 41014 Modena, Italy; m.santangelo@ausl.mo.it; 14Department of Neurology and Stroke Unit, “Morgagni-Pierantoni” Hospital, 47121 Forlì, Italy; alberto.patuelli@auslromagna.it (A.P.); marco.longoni@auslromagna.it (M.L.); 15Department of Neurology, Imola Hospital, 40026 Bologna, Italy; p.demassis@ausl.imola.bo.it; 16Department of Hospital Services, Emilia Romagna Regional Health Authority, 40127 Bologna, Italy; salvatore.ferro@regione.emilia-romagna.it

**Keywords:** amyotrophic lateral sclerosis, *C9ORF72*, clinical features, phenotype, population-based study

## Abstract

Background/Objectives: *C9ORF72* expansion is associated with significant phenotypic heterogeneity. This study aimed to characterize the clinical features of C9ALS patients from the Emilia Romagna ALS registry (ERRALS) and compare them with non-mutated ALS (nmALS) patients matched for sex, age at onset, and diagnostic delay, sourced from the same register. Methods: In total, 67 C9ALS patients were compared to 201 nmALS. Clinical data, phenotype, and prognostic factors were analyzed in the two groups and within the C9ALS group after stratification by sex. Results: C9ALS patients displayed a higher disease progression rate and shorter times to gastrostomy and invasive ventilation, despite no differences in overall survival. Female C9ALS had a more severe bulbar and upper motor neuron involvement compared to males. Cognitive and behavioral symptoms were more common in the C9ALS group, and the former was an independent prognostic factor. Prevalences of, autoimmune diseases, and dyslipidemia were significantly higher among C9ALS patients. Conclusions: In our dataset, we show an overall increased disease progression rate in C9ALS patients and hint at sex-specific discrepancies in some phenotypical characteristics. We also suggest a possible clinically relevant involvement of *C9ORF72* expansion in metabolism and autoimmunity.

## 1. Introduction

Hexanucleotide (G4C2)n repeat expansions (HREs) in the non-coding region of chromosome 9 open reading frame 72 (*C9ORF72*) are responsible for 30–50% of familial amyotrophic lateral sclerosis (fALS) and 7–10% of sporadic ALS (sALS) cases [1], as well as 5–10% of frontotemporal dementia (FTD) cases [2].

In healthy individuals, *C9ORF72* typically contains eleven or fewer hexanucleotide repeats, whereas ALS patients may present with hundreds to thousands of repeats. Although a clear pathological threshold has yet to be defined, most studies use a cutoff of 30 repeats as a reference [3].

*C9ORF72* mutations are inherited in an autosomal dominant pattern, with incomplete and age-dependent penetrance: the likelihood of developing symptoms rises from 50% at 58 years of age to 99.5% by age 83 [4].

Clinically, the *C9ORF72*-associated phenotype is highly variable. ALS associated with *C9ORF72* HRE (C9ALS) is more frequently characterized by a bulbar onset manifesting as dysphagia and dysarthria, with a higher incidence compared to ALS cases overall (30–40% vs. 25–30%, respectively) [5]. Cognitive involvement is also more common in C9ALS patients, with 20% of patients showing cognitive impairment, 10% exhibiting behavioral symptoms, and 20% meeting the criteria for the diagnosis of FTD [5].

Psychiatric symptoms are also prominent in patients with *C9ORF72* HRE, with 20–60% manifesting psychotic signs such as delusions and hallucinations, but also obsessive/compulsive disorder and catatonia [6].

Parkinsonism is observed in up to 60% of patients and can sometimes be the initial symptom, with motor neuron or cognitive deficits emerging years later [7].

Notably, around 5% of patients with a clinical presentation resembling Huntington’s disease but without the CAG expansion in *HTT* gene harbor an expansion in *C9ORF72* [7].

Despite extensive research, the pathomechanism behind the clinical heterogeneity of *C9ORF72*-associated diseases remains poorly understood. Some studies have attempted to correlate disease severity and age of onset with HRE length, with inconsistent results [8]. Somatic mosaicism, where the number of HRE varies among different tissues within the same individual, further complicates the genetic landscape. For example, the repeat size in blood can differ from that in the brain [9], which has implications for both genetic testing and the resulting disease phenotype.

To better understand the role of genetics in C9ALS, given that the spatial–temporal combination of motor and cognitive events leading to ALS onset and progression is influenced by factors such as age, sex [10], and gene variants [11], we conducted a retrospective case–control study. We thoroughly characterized phenotypically the C9ALS patients from the Italian Emilia-Romagna region’s ALS (ERRALS) registry and compared them with age- and sex-matched non-mutated ALS patients from the same registry.

## 2. Methods

### 2.1. Patients’ Data Collection

A total of 67 patients from the ERRALS registry [12], with a diagnosis of ALS according to El Escorial revised criteria [13] between 2009 and 2024, carrying an HRE in *C9ORF72* gene, were included and matched to ALS patients from the same register with no mutation in the four major ALS-related genes (*SOD1*, *FUS*, *TARDBP*, and *C9ORF72*), in order to avoid the potential effect of the other mutations on disease clinical features and progression, for a total of 201 non-mutated ALS patients (nmALS). Matched nmALS patients were selected among patients from the same registry of the same sex, age at onset (+/− 3 years), and diagnostic delay (i.e., the time between the onset of symptoms and the diagnosis (+/− 90 days)). In the case of multiple matches, nmALS patients diagnosed in the same period as C9ALS were chosen, as described in the flowchart (Figure 1). *C9ORF72* status was determined by repeat primed PCR, using the AmplideX^®^ PCR/CE C9orf72 Kit (Asuragen Inc., Austin, TX, USA), which allows to precisely quantify HRE’s lengths up to approximately 145 repeats, above which the expansion is accurately detected but not precisely quantified. The other mutations were identified using a panel that included up to 78 ALS-associated genes, as previously described [14,15].

Clinical data from all patients were collected at diagnosis and over disease course, as previously reported [12], including demographics, age at onset and diagnosis, site and time of onset, phenotype (classified as classic, bulbar, upper motor neuron (UMN) predominant, flail arm, flail leg, respiratory) [16], clinical signs such as spasticity, pathological reflexes, clonus, cramps, cognitive and/or behavioral involvement according to Strong’s criteria [17], comorbidities, drug history (including Riluzole) [18], familial history of neurodegenerative diseases (ALS, FTD, Parkinson’s disease, Alzheimer’s disease), weight, body mass index (BMI) and forced vital capacity (FVC) assessed by spirometry, time to generalization [19], disease progression using ALSFRS-r scale and disease progression rate, measured considering ALSFRS-r at diagnosis and at the last follow-up visit [20]. Data regarding the need for non-invasive ventilation (NIV), invasive ventilation (IV), enteral feeding through percutaneous endoscopic gastrostomy (PEG), and date, place, and cause of death were also gathered [21].

The disease progression rate at diagnosis was calculated as follows:$$Progression\;rate\;at\;diagnosis=\frac{\left(48-ALSFRSr\;at\;diagnosis\right)}{months\;from\;disease\;onset\;to\;diagnosis}$$

Absolute weight loss at diagnosis was defined as the difference in kilograms between the body weight during healthy status and the time of diagnosis, while relative weight loss was calculated as the percentage of the healthy weight that was lost at the time of diagnosis.

Data regarding single clinical manifestations and compound UMN and lower motor neuron (LMN) scores were also collected and quantified by the Penn Upper Motor Neuron Score (PUNMS) [22] and Devine Lower Motor Neuron Score (DLMNS) [23].

### 2.2. Statistical Analysis

We assessed differences across ALS patients’ groups by using a *t* test, ANOVA, or Chi-square tests as appropriate.

Regression analyses were conducted to evaluate the influence of clinical features on the disease progression rate.

Survival analysis was conducted using Kaplan–Meier curves, and the Log-rank test was applied for univariate analyses, while multivariate analyses were performed using the Cox regression model (by stepwise backward method).

Data analysis was performed using the STATA statistical package 15 (StataCorp. 2017. College Station, TX, USA: StataCorp LLC).

## 3. Results

Out of over 1028 patients included in the Emilia Romagna Register for ALS and tested for at least the four main genes related to ALS, 67 (6.52%) showed *C9ORF72* expansion, whereas 894 (86.96%) did not show any further mutation in *SOD1*, *FUS*, or *TARDBP* genes.

The 201 patients that best matched the 67 *C9ORF72* carriers were selected first based on diagnostic delay (±90 days) and then on sex and age at onset (±3 years).

Among the 67 C9ALS patients, 7 (10.4%) had an HRE length shorter than 145 repeats, which could be precisely quantified. In contrast, in the remaining 60 patients (89.6%), the expansion exceeded 145 repeats and could not be precisely determined (see Methods Section).

Within the nmALS cohort, some patients carried variants of uncertain significance (VUS) in the following genes: *TBK1* (n = 1), *CHMP2B* (n = 1), *DCTN1* (n = 1), *KIF5A* (n = 1), *MAPT* (n = 3), *FIG4* (n = 1), and *SQSTM1* (n = 1).

### 3.1. Demographic and Clinical Features of ALS Patients with and Without C9ORF72 Expansion

General features of the two groups are displayed in Table 1. The male/female ratio was 0.86 (31 males and 36 females among C9ALS and 93 males and 108 females among nmALS).

Family history for ALS and other neurodegenerative diseases was significantly more frequent in C9ALS patients as well as familiarity for psychiatric diseases (Table 1). Both first- and second-degree relatives for neurodegenerative diseases were more frequent in the C9ALS patients.

There were no differences in weight, BMI, and absolute and relative weight loss at diagnosis. Despite no significant difference being found in the ALSFRS-r total score at diagnosis, the bulbar sub-score was significantly lower in the C9ALS group (mean score 9.45 ± 2.42 in C9ALS group vs. 10.56 ± 2.24 in nmALS group, *p* = 0.050). FVC at diagnosis was slightly lower for the C9ALS group (89.18 ± 19.34 vs. 95.65 ± 21.86, *p* = 0.057). The time to gastrostomy and to IV was significantly shorter in C9ALS patients. Coherently, the progression rate was slower in nmALS both at diagnosis and at the last observation.

Among C9ALS patients, there were no phenotypic differences between those with more than 145 repeats and those with fewer.

### 3.2. Phenotype of ALS Patients with and Without C9ORF72 Expansion

Bulbar onset and phenotype were more frequent in the C9ALS group compared to the nmALS group (Table 2).

The compound clinical scores of upper and lower motor neuron involvements such as the DLMNS and the PUMNS did not show any difference between the two groups.

When looking at clinical symptoms, no significant difference was found in all of the clinical symptoms investigated, except for cramps, which were significantly less frequent in the C9ALS group (Table 2).

### 3.3. Cognitive Involvement of ALS Patients with and Without C9ORF72 Expansion

Cognitive and behavioral involvement was significantly more frequent in C9ALS patients compared to the nmALS group (Table 3). Following Strong’s criteria, 33.33% C9ALS patients could be classified with ALS-associated behavioral impairment (biALS) and 29.82% with ALS-associated cognitive impairment (ciALS), and 27.27% met the diagnostic criteria for FTD.

### 3.4. Comorbidities of ALS Patients with and Without C9ORF72 Expansion

Autoimmune diseases and dyslipidemia were significantly more frequent in the C9ALS group compared to nmALS patients (13.43% vs. 3.48%, *p* = 0.003 and 38.30% vs. 16.40%, *p* = 0.001, respectively). No significant difference was found in other comorbidities between C9ALS and nmALS patients except for depression, which was more frequent in the second group (21.21 vs. 39.74, *p* = 0.017) (Table 4).

### 3.5. Progression Rate and Survival of ALS Patients with and Without C9ORF72 Expansion

Regression analysis confirmed that the presence of *C9ORF72* expansion (Coef: 0.45, 95% CI: 0.04 to 0.87, *p* = 0.033), as well as a younger age at onset (Coef: −0.03, 95% CI: −0.05 to −0.007, *p* = 0.008), led to a faster disease progression rate.

No difference was found in overall survival between the two groups (33.48 ± 18.85 vs. 39.42 ± 28.12, *p* = 0.199) (Figure 2).

Univariate Cox regression analysis and multivariate analysis for patients with C9ALS are shown in Table 5. In C9ALS patients, the multivariate analysis of survival showed that independent prognostic factors for tracheostomy-free survival were diagnostic delay (HR = 0.92, 95% CI 0.86–0.98, *p* = 0.014), disease progression rate at diagnosis (HR = 1.65, 95% CI 1.10–2.47, *p* = 0.016), and presence of cognitive involvement (HR = 7.70 95% CI 3.12–19.02, *p* < 0.001).

### 3.6. Sex-Related Differences in C9ALS Patients

When comparing clinical differences between sexes within the nmALS and the C9ALS groups (Table 6), no significant sex-related differences were found in the main clinical features, except for diagnostic delay, which was significantly shorter in male nmALS patients.

A higher disease progression rate at diagnosis was found for C9ALS males, despite not reaching clinical significance; this trend, albeit not significant, was confirmed also for time from symptoms onset to tracheostomy and death.

The site of onset was more frequently bulbar in the female population in both groups, and the ALSFRS-r bulbar subscale score was significantly higher in male C9ALS patients (10.16 ± 2.08 vs. 8.87 ± 2.56, *p* = 0.047).

While no differences were found in DLMNS, the PUMNS resulted in being significantly higher in females compared to males (9.92 ± 5.75 vs. 5.53 ± 5.12, *p* = 0.004).

Finally, when considering comorbidities, autoimmune diseases and psychosis were more frequent in female C9ALS patients, despite not reaching statistical significance

Despite a tendency towards a higher disease progression rate, shorter time to IV, or death in male C9ALS patients, no statistically significant difference in survival was found between men and women (Figure 3).

When analyzing prognostic factors in the C9ALS population stratified by sex, we found that cognitive changes (HR 3.87, 95% CI 1.12–13.34, *p* = 0.032), disease progression rate at diagnosis (HR 5.06, 95% CI 2.05–12.48, *p* < 0.001), and concomitant psychosis (HR 8.23, 95% CI 1.12–60.75, *p* = 0.039) were independent prognostic factors in women, whereas weight loss at diagnosis was the only independent prognostic variable in men (HR 1.16, 95% CI 1.02–1.33, *p* = 0.023) (Appendix A).

## 4. Discussion

In this study, we explore the phenotypic heterogeneity of ALS associated with the *C9ORF72* mutation [5], comparing a population-based cohort of C9ALS patients with nmALS patients matched for sex, age, and diagnostic delay, sourced from the same population-based register.

As expected [24], we observed that both bulbar onset and bulbar phenotype were more frequent in the C9ALS cohort compared to the nmALS group. This was further corroborated by a significantly lower ALSFRS-r bulbar subscale score at diagnosis in the C9ALS group. However, unlike previous studies [25], bulbar onset in our C9ALS cohort was not associated with shorter survival. Interestingly, we identified a sex-related difference in bulbar involvement at diagnosis, with the ALSFRS-r bulbar subscale being lower in women, showing consistency with prior findings [11,25,26]. The mechanism behind the more frequent bulbar involvement in C9ALS is poorly understood, but some studies have identified multiple molecular subtypes in ALS patients which differed between bulbar and spinal onset patients, suggesting that different motor neurons could be susceptible to different pathological mechanisms [27]. For example, bulbar motor neurons may be more vulnerable to the pathological mechanisms that characterize C9ALS but are absent in nmALS, such as RNA foci formation, DPR accumulation, and C9orf72 loss of function [1].

We analyzed UMN and LMN involvement using two compound scores, PUMNS and DLMNS, respectively, in both C9ALS and nmALS cohorts. While no significant differences in global UMN and LMN scores were observed between the two groups, the stratification of C9ALS patients by sex revealed significantly higher PUMNS scores in females, indicating more prominent UMN involvement in this subgroup. No sex-related differences in PUMNS were found among nmALS patients, suggesting that increased UMN involvement may specifically affect female C9ALS patients. These findings point to the existence of sex-related differences in C9ALS pathology.

Family history of ALS was significantly more prevalent in C9ALS patients (up to 43.93%) compared to nmALS patients, consistent with *C9ORF72* being the most common genetic cause of familial ALS [1]. Furthermore, the prevalence of family history for all neurodegenerative diseases in the C9ALS cohort reached 81.82%, underscoring the broader role of *C9ORF72* in neurodegeneration [8]. Supporting this hypothesis, evidence suggests that C9orf72 regulates microglial amyloid clearance [28], potentially linking its dysfunction to broader neurodegenerative processes.

Indeed, the clinical spectrum of *C9ORF72* HRE includes movement disorders and FTD [7]. It is plausible that, in some cases, Parkinsonism and dementia represent manifestations of *C9ORF72* pathology without overt motor neuron involvement. In our cohort, only a few patients presented with Parkinsonism or psychosis, with no significant differences between the two groups. Depression, however, was significantly less frequent in the C9ALS cohort, possibly reflecting a lack of insight associated with cognitive involvement [29]. A family history of psychiatric diseases was significantly more frequent in C9ALS patients, supporting the role of *C9ORF72* in psychiatric disease development [30].

Interestingly, psychiatric disorders were more common in females compared to males, suggesting a potential role for additional factors, such as hormonal influences, in the neuropsychiatric manifestations of C9ALS. Psychiatric involvement also emerged as an independent prognostic variable in female C9ALS patients. Although the association between C9ALS and psychiatric symptoms is well established [30], their impact on prognosis remains underexplored. The association between worst progression and psychiatric involvement, which is frequently seen in FTD [31], may simply reflect the poorer outcomes of the association of FTD in C9ALS patients [32].

Moreover, in our study, we observed a significantly higher incidence of autoimmune diseases in the C9ALS cohort, a result in line with the biological data on autoimmune phenotype in *C9ORF72* knockout mouse models [33,34]. Despite the established relation between loss of function of C9orf72 and autoimmunity, only one prior study has reported an increased incidence of autoimmune diseases in carriers of the *C9ORF72* HRE [35]. In our cohort, autoimmune diseases were more prevalent in female C9ALS patients than in males, though this difference did not reach statistical significance. While this observation might simply reflect the generally higher incidence of autoimmune diseases in females [36], emerging evidence highlights a critical role of *C9ORF72* in immunity [37,38]. Therefore, a potential contribution of *C9ORF72* HRE to autoimmunity in females cannot be ruled out.

Furthermore, the higher prevalence of dyslipidemia observed in female C9ALS patients compared to males suggests a complex interplay between genetic and hormonal factors in modulating metabolism [39]. The relationship between dyslipidemia and systemic inflammation is well documented [40] and may reflect metabolic dysfunction driven by the *C9ORF72* mutation. This aligns with the mutation’s known involvement in immune regulation and inflammatory processes [41]. These findings, together with the increased incidence of autoimmune diseases, support the hypothesis of a broader role for C9orf72 in regulating metabolic and immune homeostasis [42]. A deeper understanding of the role of C9orf72 in metabolism and immunity could help identify specific molecular signatures for C9ALS, potentially enabling personalized treatment strategies. This approach has been recently suggested in studies investigating ALS-specific signatures in blood and CSF [43,44].

C9ALS patients are typically reported to have shorter survival, but prior studies on this topic have yielded conflicting results [25,45]. In our cohort, we found a significantly higher disease progression rate in C9ALS patients at both diagnosis and last observation. These patients also exhibited significantly shorter times to PEG and tracheostomy, consistent with the earlier and more frequent bulbar involvement in this cohort. Some studies have attempted to correlate the HRE length with disease progression and survival, but the findings remain inconclusive [8,45,46].

Unfortunately, since genetic testing was performed in a clinical setting, we do not have the exact number of *C9ORF72* repeat expansions when the expansion exceeds a threshold of 145 repeats, which is widely recognized as above the pathological cutoff [7]. Precise sizing for very large expansions in fact requires complementary methods such as Southern blot analysis, which is not commonly used in clinical practice.

As a result, for most patients, data on the precise length of the HRE are unavailable, representing a significant limitation of our study. When analyzing potential correlations between the HRE length and key clinical variables, we found no significant differences between C9ALS patients with >145 and < 145 repeats. However, this could be due to the limited size of our sample.

Although not reaching statistical significance, our data showed reduced respiratory function at diagnosis in C9ALS patients. This observation might suggest the early subclinical involvement of respiratory muscles, particularly the phrenic motor neurons, in the initial stages of the disease [47]. Supporting this, we observed a significantly shorter time from diagnosis to tracheostomy in C9ALS patients. Previous studies have also reported increased vulnerability of phrenic MNs in *C9ORF72*-mutated iPSC models [47].

We did not find, however, a significant difference in overall survival, probably because of the small size of our cohort.

Multivariate analysis in our cohort of C9ALS patients highlighted the disease progression rate at diagnosis, time to diagnosis, and cognitive involvement classified as ALSci, as independent prognostic variables, consistent with previous findings [32].

Finally, we confirm that females with *C9ORF72* expansions are more likely to present with a bulbar phenotype independent of age and are more prone to developing neuropsychiatric symptoms [25,31]. Sex differences in C9ALS patients’ disease progression were also evident, with males generally experiencing a more aggressive course and shorter survival times [10]. This is in line with the known hormonal influences, in particular the neuroprotective effects of estrogen, which may contribute to attenuating disease progression in females [48]. While the pathological hallmark of *C9ORF72*-associated diseases—TDP-43 proteinopathy—appears similar in males and females, subtle differences in its regional distribution or severity might exist and contribute to the reported differences. Further studies will be needed to address those discrepancies [49].

Further research to document sex differences in *C9ORF72* expansions could have important implications for developing tailored therapeutic strategies in the future.

## 5. Conclusions

This study highlights the phenotypic heterogeneity of *C9ORF72*-associated ALS, revealing significant clinical and sex-related differences compared to non-mutated ALS. A family history of ALS and neurodegenerative diseases was significantly more common in C9ALS, supporting its broad role in neurodegeneration. Bulbar involvement was more frequent and severe in C9ALS, with women showing greater upper motor neuron involvement and more pronounced bulbar dysfunction. Psychiatric symptoms, particularly in females, emerged as a distinct feature and independent prognostic factor, potentially influenced by hormonal and genetic factors.

C9ALS patients demonstrated faster disease progression and an earlier need for interventions, though overall survival did not differ from nmALS, possibly due to the limited cohort size.

Sex differences in disease progression, psychiatric and autoimmune involvement, and bulbar phenotypes emphasize the importance of tailoring therapeutic strategies. Future research should focus on the interplay between genetic, immune, and sex-specific factors to better understand and manage C9ALS.

## Figures and Tables

**Figure 1 genes-16-00309-f001:**
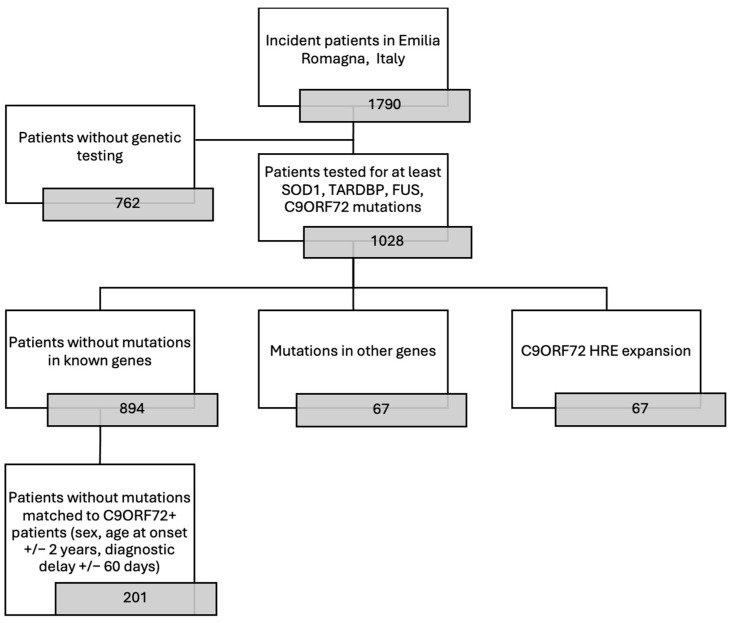
Flowchart showing patient selection.

**Figure 2 genes-16-00309-f002:**
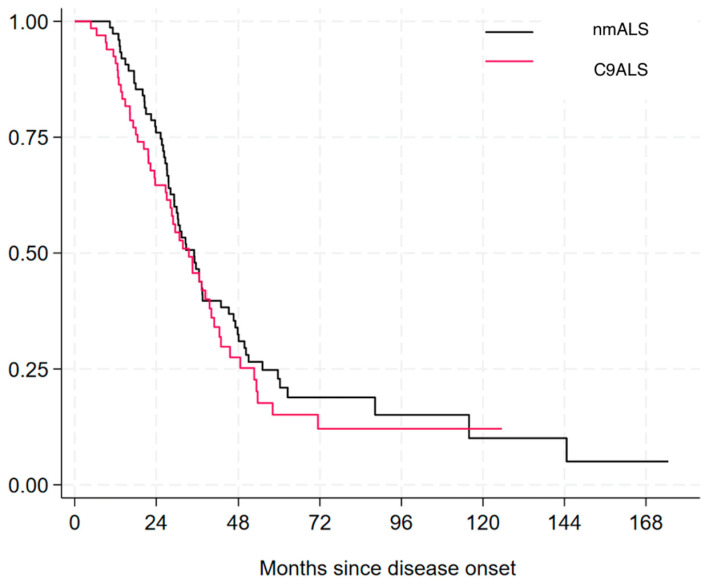
Kaplan–Meier showing survival in C9ALS and nmALS patients.

**Figure 3 genes-16-00309-f003:**
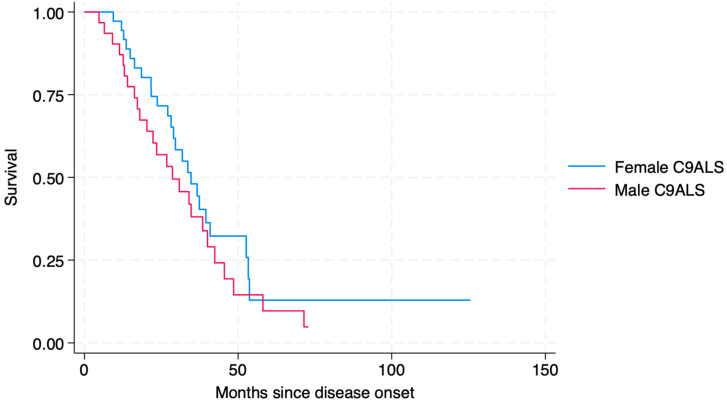
Kaplan–Meier showing survival in male and female C9ALS patients.

**Table 1 genes-16-00309-t001:** Clinical data from ALS patients with *C9ORF72* mutations (C9ALS) and patients without mutations in the four major genes (nmALS). BMI: body mass index; ALSFRS-R: ALS Functional Rating Scale Revised; FVC: forced vital capacity; NIV: non-invasive ventilation; PEG: percutaneous endoscopic gastrostomy; IV: invasive ventilation.

	C9ALS, n (%), Mean [SD]	NmALS, n (%), Mean [SD]	*p*-Value
Age at onset, years	57.63 [9.41]	57.88 [9.01]	0.845
Diagnostic delay, months	9.89 [7.11]	9.76 [6.34]	0.889
Family history for ALS ^1^	29 (43.93)	7 (3.57)	**<0.001**
Family history for dementia ^2^	25 (37.88)	42 (21.21)	**0.009**
Family history for other neurodegenerative diseases ^2^	44 (66.67%)	41 (20.92%)	**<0.001**
Family history for psychiatric disease ^3^	8 (4.98%)	5 (2.55%)	**0.002**
Family ^4^ history among 1st-degree relatives	46 (76.79%)	21 (27.27%)	**<0.001**
Family ^4^ history among 2nd-degree relatives	19 (33.33%)	11 (14.60%)	**0.013**
BMI at diagnosis, kg/m^2^	24.06 [4.14]	24.39 [4.14]	0.592
Weight loss at diagnosis (kg)	3.51 [6.38]	3.01 [5.74]	0.584
Weight loss at diagnosis (%) ^5^	4.68 [9.25]	3.86 [7.37]	0.493
ALSFRS-r at diagnosis, points ^6^	40.08 [4.37]	41.10 [6.00]	0.202
Disease progression rate at diagnosis (points/month) ^7^	1.53 [1.50]	1.06 [1.34]	**0.029**
ALSFRS-r at last observation, points ^8^	19.60 [11.38]	21.14 [12.73]	0.389
Disease progression rate at last observation (points/month) ^9^	1.60 [1.34]	1.01 [1.58]	**0.016**
FVC at diagnosis (%) ^10^	89.18 [19.34]	95.65 [21.86]	0.057
NIV	30 (4478)	98 (48.76)	0.572
Time to NIV, months	22.72 [11.91]	28.09 [21.75]	0.207
PEG	32 (47.76)	85 (42.29)	0.434
Time to PEG, months	24.57 [12.60]	33.73 [24.58]	**0.050**
IV	18 (26.87)	48 (23.88)	0.623
Time to IV, months	26.87 [21.17]	38.69 [20.22]	**0.041**
Time to death, months	33.48 [18.85]	39.42 [28.12]	0.199
Total	67 (100)	201 (100)	

^1^ Data on the presence of family history for ALS were available for 66 out of 67 C9ALS and 196 out of 201 nmALS. ^2^ Data on the presence of family history for dementia and other neurodegenerative diseases were available for 66 out of 67 C9ALS and 198 out of 201 nmALS. ^3^ Data on the presence of family history for psychiatric diseases were available for 65 out of 67 C9ALS and 196 out of 201 nmALS. ^4^ Data on the degree of kinship were available for 57 out of 67 C9ALS and 74 out of 201 nmALS. ^5^ Data on weight loss were available for 57 out of 67 C9ALS and 179 out of 201 nmALS. ^6^ Data on ALSFRS-r at diagnosis were available for 66 out of 67 C9ALS and 193 out of 201 nmALS. ^7^ Data on disease progression rate at diagnosis were available for 54 out of 67 C9ALS and 183 out of 201 nmALS. ^8^ Data on ALSFRS-r at last observation were available for 62 out of 67 C9ALS and 201 out of 201 nmALS. ^9^ Data on disease progression rate at last observation were available for 51 out of 67 C9ALS and 188 out of 201 nmALS. ^10^ Data on FVC at diagnosis were available for 58 out of 67 C9ALS and 117 out of 201 nmALS. Statistically significant values are displayed in bold.

**Table 2 genes-16-00309-t002:** Disease onset, clinical phenotypes, clinical signs, and compound scores of UMN and LMN involvement in C9ALS and nmALS. UMN: upper motor neuron; LMN: lower motor neuron.

	C9ALS n (%), Mean [SD]	nmALS n (%), Mean [SD]	*p*-Value
Site of onset			
Bulbar	23 (34.33)	52 (28.42)	0.436
Spinal	44 (65.67)	130 (71.04)	0.440
Respiratory	0 (0.00)	1 (0.55)	1.000
Phenotype ^1^			
Flail	6 (8.96)	32 (16.58)	0.162
UMN predominant	2 (2.99)	5 (2.59)	1.000
Bulbar	23 (34.33)	46 (23.83)	0.109
Respiratory	0 (0.00)	1 (0.52)	1.000
Classic	36 (55.77)	109 (56.48)	0.776
UMN and LMN involvement ^2^			
Penn Upper Neuron Motor Score (0–32)	3.82 [2.44]	3.45 [3.12]	0.421
Devine Lower Motor Neuron Score (0–12)	7.52 [5.80]	7.78 [5.93]	0.788
Spasticity in most affected limb (Ashworth 0–2)	58 (95.09)	124 (91.85)	0.600
Spasticity in most affected limb (Ashworth 3–4)	3 (4.91)	11 (8.15)	
Palmomental reflex	11 (19.64)	33 (24.81)	0.443
Glabellar reflex	7 (10.50)	14 (10.69)	0.719
Snout reflex	9 (16.07)	26 (19.40)	0.589
Masseter reflex	18 (32.14)	27 (20.45)	0.086
Hoffman reflex	8 (13.56)	20 (14.81)	0.071
Babinski reflex	6 (10.00)	15 (11.11)	0.879
Clonus	5 (8.47)	15 (11.19)	0.594
Cramps	11 (18.33)	48 (49.48)	**<0.001**

^1^ Data on phenotype were available for 67 out of 67 C9ALS and 193 out of 201 nmALS. ^2^ Data on UMN and LMN were available for 41 out of 67 C9ALS and 130 out of 201 nmALS. Statistically significant values are displayed in bold.

**Table 3 genes-16-00309-t003:** Cognitive and behavioral involvement in C9ALS and nmALS patients. ALSbi: ALS-associated behavioral impairment; ALSci: ALS-associated cognitive impairment; FTD: frontotemporal dementia.

	C9ALS, n (%)	nmALS, n (%)	*p*-Value
ALSbi ^1^	19 (33.33)	13 (9.56)	**<0.001**
ALSci ^1^	17 (29.82)	9 (6.62)	**<0.001**
ALS-FTD ^2^	18 (27.27)	9 (6.57)	**<0.001**
Pseudobulbar syndrome ^1^	18 (31.03)	28 (20.59)	0.117

^1^ Data on presence of ALSbi, ALSci, and pseudobulbar syndrome were available for 57 out of 67 C9ALS and 136 out of 201 nmALS. ^2^ Data on presence of ALS-FTD were available for 66 out of 67 C9ALS and 137 out of 201 nmALS. Statistically significant values are displayed in bold.

**Table 4 genes-16-00309-t004:** Comorbidities in C9ALS and nmALS patients.

	C9ALS, n (%)	nmALS, n (%)	*p*-Value
Dyslipidemia ^1^	18 (38,30)	33 (16.40)	**0.001**
Autoimmune diseases ^1^	9 (13.43)	7 (3.48)	**0.003**
Chronic Obstructive Pulmonary Disease (COPD)	2 (2.99)	7 (3.48)	0.845
Respiratory diseases (excluding COPD)	2 (2.99)	6 (2.99)	0.470
Diabetes ^2^	2 (3.03)	6 (2.99)	0.985
Hypertension	21 (31.34)	58 (28.86)	0.699
Cardiopathies	7 (10.45)	17 (8.46)	0.621
Parkinsonism ^2^	2 (3.08)	3 (1.49)	0.207
Cancer history ^3^	8(11.94)	8 (10.26)	0.747
Psychosis ^4^	3 (4.62)	3 (1.49)	0.140
Depression ^5^	14 (21.21)	31 (39.74)	**0.017**

^1^ Data on dyslipidemia were available for 47 out of 67 C9ALS and 201 out of 201 nmALS. ^2^ Data on diabetes and Parkinsonism were available for 66 out of 67 C9ALS and 201 out of 201 nmALS. ^3^ Data on cancer history were available for 67 out of 67 C9ALS and 78 out of 201 nmALS. ^4^ Data on presence of psychosis were available for 65 out of 67 C9ALS and 201 out of 201 nmALS. ^5^ Data on presence of depression were available for 66 out of 67 C9ALS and 201 out of 201 nmALS. Statistically significant values are displayed in bold.

**Table 5 genes-16-00309-t005:** Univariate Cox regression analysis and multivariate analysis of survival in C9ALS patients. BMI: body mass index; FVC: forced vital capacity; ALSbi: ALS-associated behavioral impairment; ALSci: ALS-associated cognitive impairment; FTD: frontotemporal dementia. Statistically significant values are displayed in bold.

	Univariate		Multivariate	
Variable	HR (95% CI)	*p*-Value	HR (95% CI)	*p*-Value
Sex, male	1.49 (0.85–2.61)	0.165		
Family history, presence	0.94 (0.53–1.66)	0.821		
Age at onset, years	1.01 (0.98–1.04)	0.466		
Diagnostic delay, months	0.92 (0.87–0.97)	0.004	0.92 (0.86–0.98)	**0.014**
Time to generalization, months	0.96 (0.92–0.99)	0.008		
BMI at diagnosis, kg/m^2^	1.04 (0.99–1.10)	0.131		
FVC at diagnosis, %	0.99 (0.97–1.00)	0.156		
ALSFRS-r at diagnosis, points	0.98 (0.92–1.04)	0.472		
Disease progression rate at diagnosis, points/month	1.40 (1.16–1.70)	0.001	1.65 (1.10–2.47)	**0.016**
Weight loss, % of healthy weight	1.07 (1.02–1.11)	0.003		
Onset, bulbar	0.99 (0.55–1.77)	0.965		
ALSci, presence	2.85 (1.48–5.46)	0.002	7.70 (3.12–19.02)	**<0.001**
ALSbi, presence	2.33 (1.23–4.42)	0.009		
ALS-FTD, presence	2.38 (1.31–4.33)	0.004		
Depression, presence	0.65 (0.31–1.35)	0.248		
Psychosis, presence	3.16 (0.96–10.39)	0.058		
Chronic Obstructive Pulmonary Disease (COPD), presence	3.89 (0.90, 16.91)	0.070		
Other respiratory diseases, presence	14.00 (2.78, 70.44)	0.001		
Diabetes, presence	1.02 (0.25–4.24)	0.978		
Cardiopathies, presence	1.59 (0.67–3.76)	0.294		
Hypertension, presence	1.39 (0.77–2.51)	0.276		
Dyslipidemia, presence	1.00 (0.51–1.94)	0.993		
Autoimmune diseases, presence	0.94 (0.37–2.40)	0.903		
Cancer history, presence	1.10 (0.47–2.60)	0.823		
Previous trauma, presence	1.24 (0.62–2.48)	0.540		
Former tobacco smoking	1.14 (0.58–2.25)	0.701		
Current tobacco smoking	0.74 (0.26–2.11)	0.569		

**Table 6 genes-16-00309-t006:** Clinical data, site of onset, clinical phenotype, and comorbidities in male and female C9ALS patients. Statistically significant values are displayed in bold.

	nmALS			C9ALS		
	Women, n (%), Mean [SD]	Men, n (%), Mean [SD]	*p*-Value	Women, n (%), Mean [SD]	Men, n (%), Mean [SD]	*p*-Value
Age at onset, years	58.57 [8.29]	57.12 [9.73]	0.256	57,88 [9.08]	57.34 [1.78]	0.817
Diagnostic delay, months	10.89 [7.19]	8.53 [5.01]	**0.008**	10.89 [8.24]	8.74 [5.41]	0.219
BMI at diagnosis, kg/m^2^	23.95 [4.68]	24.89 [3.41]	0.134	23.83 [5.47]	24.33 [3.37]	0.674
Weight loss at diagnosis (Kg)	2.43 [5.61]	3.67 [5.86]	0.161	2.32 [6.58]	5.02 [5.90]	0.943
Weight loss at diagnosis (%)	3.29 [7.90]	4.50 [6.71]	0.285	3.54 [10.73]	6.13 [6.86]	0.299
ALSFRS-r at diagnosis, points	40.74 [6.26]	41.52 [5.70]	0.368	39.2 [4.56]	41.06 [3.98]	0.083
Disease progression rate at diagnosis (points/month)	0.91 [1.16]	1.21 [1.51]	0.137	1.29 [1.12]	1.82 [1.85]	0.200
FVC at diagnosis, %	95.54 [24.10]	95.78 [19.24]	0.952	89.37 [19.32]	88.94 [19.78]	0.933
Time to NIV, months	28.34 [20.08]	27.86 [23.48]	0.915	20.73 [9.93]	24.87 [13.77]	0.358
Time to PEG, months	33.54 [25.07]	33.94 [24.36]	0.940	24.59 [11.73]	24.56 [14.04]	0.995
Time to tracheostomy, months	37.67 [22.17]	39.32 [19.30]	0.789	37.33 [26.02]	20.21 [15.18]	0.095
Time to death, months	36.15 [26.07]	41.46 [29.49]	0.482	32.09 [25.35]	35.09 [20.70]	0.572
Time to death/last observation, months	42.31 [32.62]	41.72 [26.58]	0.932	36.47 [23.53]	30.00 [18.07]	0.222
ALSFRS-r at last observation, points	20.69 [12.86]	21.64 [12.64]	0.598	18.41 [9.97]	21.00 [13.04]	0.398
Time from diagnosis to last observation, months	36.20 [37.18]	38.10 [32.92]	0.703	18.86 [12.21]	18.50 [14.43]	0.917
Disease progression rate at last observation, (points/month)	1.11 [1.91]	0.90 [1.13]	0.368	1.39 [1.04]	1.89 [1.67]	0.196
Site of onset, bulbar	34 (32.38)	19 (19.79)	**0.043**	15 (41.67)	8 (25.81)	0.173
DLMNS (0–12)	4.00 [2.67]	3.65 [2.20]	0.419	3.28 [3.26]	3.66 [3.02]	0.681
PUMNS (0–32)	8.10 [5.55]	7.45 [6.32]	0.519	9.92 [5.76]	5.53 [5.12]	**0.004**
ALSFRS-r bulbar score at diagnosis	10.33 [2.26]	10.82 [2.21]	0.153	8.87 [2.56]	10.16 [2.08]	**0.047**
ALSFRS-r upper limb score at diagnosis	13.13 [3.36]	12.65 [3.66]	0.368	13.10 [3.06]	13.76 [2.30]	0.377
ALSFRS-r upper + lower limb score at diagnosis	18.87 [4.91]	18.41 [4.98]	0.543	18.97 [4.63]	19.44 [3.74]	0.683
ALSFRS-r lower limb score at diagnosis	5.74 [2.23]	5.76 [2.22]	0.946	5.87 [2.29]	5.68 [2.14]	0.757
ALSFRS-r respiratory score at diagnosis	11.63 [1.09]	11.58 [1.25]	0.806	11.67 [0.71]	11.84 [0.47]	0.303
ALSbi	3 (4.55)	10 (14.29)	0.054	12 (37.50)	7 (28.00)	0.450
ALSci	4 (6.06)	5 (7.14)	0.800	9 (28.12)	8 (32.00)	0.751
ALS-FTD	3 (4.48)	6 (8.57)	0.334	11 (30.56)	7 (23.33)	0.512
Dyslipidemia	18 (17.14)	15 (15.62)	0.772	7 (29.17)	11 (47.83)	0.188
Autoimmune diseases	3 (2.86)	4 (4.17)	0.613	7 (20.59)	2 (6.06)	0.081
Cancer history	5 (14.71)	3 (6.82)	0.255	5 (14.71)	3 (9.09)	0.479
Depression	15 (44.12)	16 (36.36)	0.488	10 (29.41)	4 (12.50)	0.093
Psychosis	2 (1.90)	1 (1.04)	0.614	3 (8.82)	0 (0.00)	0.090
Parkinsonism	1 (0.95)	2 (2.08)	0.509	0 (0.00)	2 (6.25)	0.145
Chronic Obstructive Pulmonary Disease	5 (4.76)	2 (2.08)	0.301	1 (2.94)	1 (3.03)	0.983
Diabetes	3 (2.86)	3 (3.12)	0.911	2 (5.88)	0 (0.00)	0.164
Hypertension	26 (24.76)	32 (33.33)	0.180	11 (32.35)	10 (30.30)	0.856
Cardiopathies	7 (6.67)	10 (10.42)	0.340	3 (8.82)	4 (12.12)	0.659

## Data Availability

All data are available upon motivated request.

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
