# Peer review of "Phenotypical Characterization of C9ALS Patients from the Emilia Romagna Registry of ALS: A Retrospective Case–Control Study"

_genes, 2025, doi:10.3390/genes16030309_

Round 1
Reviewer 1 Report
Comments and Suggestions for Authors
In this paper authors described clinical and instrumental phenotypes recorded in patients with C9orf72 mutations and in those without mutation in main ALS-associated genes (SOD1, FUS, and TARDBP). Both groups were selected from the Emilia Romagna ALS Registry (ERRALS). The study is throroughly described, and the comparisons between groups are well conducted.
Some minor and major revisions should be performed to improve the manuscript.
Minor revisions
A full revision of the paper is recommended to correct some typos and enhance the clarity of the English language. Below there are some specific suggestions:
Line 68 – “HTT gene”: genes should be reported in italic
Line 88 – “(SOD1, FUS, TARDBP and C9ORF72)”: genes should be reported in italic
Page 5 – Table 1: some captions are reported in italic, while others are not. Is there a reason for this inconsistency?
Line 180 – “camps”: the authors may have intended to write “cramps”
Line 352 – the sentence “However, these remain insufficiently studied” should be rephrased for clarity
Major revisions
The paper reports the selection process of patients (Figure 1). It is noted that 67 patients carried mutations in genes other than C9orf72 (SOD1, FUS, or TARDBP). It would be appropriate to include a note explaining why these patients were excluded from the study.
Additionally, there is currently no well-defined pathological threshold for the number of hexanucleotide repeats in C9orf72. Multiple studies have tried to establish normal and pathological ranges, revealing significant variability among populations. It would be highly interesting to include data on the distribution of HREs among C9ALS patients in ERRALS.
Comments on the Quality of English LanguageThe English language is good, but some sentences require a revision to enhance the clarity.
Reviewer 2 Report
Comments and Suggestions for Authors
1. What are the other mutations in the study cohort?
2. What was the range of the C9ORF72 repeat length? What was the longest, and what was the most miniature repeat?
3. The authors used "C9ORF72 status was determined by repeat primed PCR, as previously reported". What is the length of repeat products this PCR can detect? Is it possible that very long repeats have been missed?
4. Please describe the PCR method in detail. I checked your reference to the method, and it is not there. You must provide a complete PCR protocol for the C9ORF72 and also the protocol for genotyping the other genes.
5. The authors can actually discuss some more recent findings on the subtypes of ALS and potential genetic mechanisms and biomarkers (PMID: 38129934, 38001563, 39844875). These references would help to get a better translational overview of the main findings.
Round 2
Reviewer 2 Report
Comments and Suggestions for Authors
The authors have addressed all the comments.
Author Response
No requested reply.